# A Case Report of Zygomatic Fracture Reconstruction: Evaluation with Orbital Measurements and Models Registration

Khalil Yousof [1], Mhd Ayham Darwich [2], Khaldoun Darwich [1], Ghina Alassah [1], Ahmed Imran [3] and Hasan Mhd Nazha [4],*

[1] Department of Oral and Maxillofacial Surgery, Faculty of Dentistry, Damascus University, Damascus, Syria; khalil9.yousof@damascusuniversity.edu.sy (K.Y.); khaldoun.darwich@damascusuniversity.edu.sy (K.D.); ghina.alassah2@damascusuniversity.edu.sy (G.A.)

[2] Department of Biomechanics, Faculty of Biomedical Engineering, Al-Andalus University for Medical Sciences, Tartous, Syria; a.darwich@au.edu.sy

[3] Department of Biomedical Engineering, Ajman University, Ajman P.O. Box 346, United Arab Emirates; a.imran@ajman.ac.ae

[4] Faculty of Mechanical Engineering, Institute of Mechanics, Otto Von Guericke University Magdeburg, 39106 Magdeburg, Germany

* Correspondence: hasan.nazha@ovgu.de

**Abstract:** The repair and reconstruction of defects in the craniomaxillofacial region can be particularly challenging due to the complex anatomy, individuality of each defect, and sensitivity of the involved systems. This study aims to enhance the facial appearance and contribute to the reconstruction of the zygomatic arch. This was achieved through virtual planning of the surgery and assessment of clinical matching, including orbital measurements and registration of numerical models. A three-dimensional design of a young female case was generated on a skull model using Mimics® software, and the orbit was isolated using 3-Matic® to assess the reconstructive effect. 3D-printed implants were then surgically placed on the injured region, and Netfabb® software was used to make a virtual registration between the numerical models before and after the intervention. This allowed for the calculation of a deviation of 7 mm, equivalent to 86.23% of the shape restoration rate, to assess the success of the surgery. The computerized method enabled a precise design of the needed plates and analysis of the fixation places, resulting in a satisfactory cosmetic and functional outcome for the patient with minimal complications and good implant stability. Notably, a significant difference was observed in the orbital frontal area after 3 months of surgery ($p < 0.001$). Within the limitations of the study, these results suggest that virtual planning and customized titanium implants can serve as useful tools in the management of complex zygomatic-orbital injuries.

**Keywords:** CAD/CAM; patient-specific implant; zygomatico-orbital complex; surgical planning

## 1. Introduction

Facial injuries can significantly impact physical and psychological well-being of the patient. Effective, individualized treatment is essential to restore appearance and function and prevent disfigurement, stigma, and lowered self-esteem and social functioning [1].

One of the most important challenges facing maxillofacial surgeons is the reconstruction of the orbital region. The orbital region is a complex anatomical structure that contains vital organs, such as the eye and its associated muscles, nerves and vessels [2]. Orbital trauma can result in functional and aesthetic impairments that affect the quality of life of patients. Therefore, orbital reconstruction aims to restore the normal anatomy and physiology of the orbit by using various techniques and materials [3].

Orbital reconstruction can be performed using conventional methods or computer-assisted methods. Conventional methods rely on clinical examination, radiographic imaging, and surgical experience to guide the placement of orbital implants. However, these methods have limitations such as poor accuracy, high variability, and difficulty in assessing complex defects [4]. Computer-assisted methods use advanced technologies such as virtual surgical planning (VSP) [5], intraoperative imaging (IOI) [6], intraoperative navigation (ION) [7], and mixed reality (MR) to improve the predictability and reliability of orbital reconstruction [8]. Computer-assisted methods have shown promising results in improving the accuracy, precision, efficiency, and safety of orbital reconstruction compared to conventional methods. However, these methods also have challenges such as high costs, technical difficulties, learning curves, and ethical issues that must be addressed before they can be widely adopted in clinical practice [9,10].

In recent years, new techniques have emerged to overcome these drawbacks and improve the outcomes of orbital reconstruction. One of these techniques is 3D printing technology, which allows for the customized fabrication of implants based on patient-specific anatomy and defect size [11]. 3D printing technology can create complex and precise shapes that match the contour of the orbital wall and avoid gaps or protrusions that may compromise the aesthetic and functional results. Moreover, 3D printing technology can use biodegradable or bioactive materials that can integrate with the surrounding bone tissue and stimulate bone regeneration. Several studies have reported successful applications of 3D-printed implants for orbital reconstruction in various clinical scenarios, demonstrating their feasibility, safety, and efficacy [12,13].

Patient-specific implant (PSI) is a personalized approach to reconstructive and esthetic surgery that aims to restore the normal function and appearance of damaged or missing body parts [14]. PSI is especially useful for complex anatomical structures, such as the skull, where conventional implants may not fit well or match the patient's expectations. PSI involves using computer-aided design (CAD) and computer-aided manufacturing (CAM) techniques to create customized implants based on the patient's preoperative imaging data [14]. The implants can be made of various biocompatible materials, such as titanium, polymethyl methacrylate (PMMA) [15], polyether ether ketone (PEEK) [16], or ceramic [17]. PSI offers several advantages over standard implants, such as improved accuracy, reduced intraoperative time, better cosmetic outcomes, and lower risk of infection or complications. Furthermore, PSI fabrication not only improves the aesthetic outcome for patients with facial injuries but also enhances their health-related quality of life (HRQoL) [18]. HRQoL is a multidimensional concept that reflects how patients perceive their physical, mental, and social well-being in relation to their health condition.

The use of PSI requires advanced imaging modalities, such as high-resolution computed tomography (CT) images to design and print the implant according to each patient's unique anatomy and defect characteristics [19]. This approach allows for a more precise fit and alignment of the implant with the surrounding bone and soft tissue structures, which can improve the functional and aesthetic outcomes of the surgery. Moreover, PSI can reduce operative time and blood loss by minimizing the need for intraoperative adjustments and bone grafting. PSI also enables the creation of complex geometries and porous structures that can enhance the biomechanical stability and osseointegration of the implant [20]. However, PSI also poses some challenges such as higher costs, longer lead times, regulatory issues, and ethical concerns. Therefore, careful patient selection and multidisciplinary collaboration are essential for the successful application of PSI in orthopedic surgery [21].

Compared to previous studies performed in the same context, this study introduces a novel approach to reconstructing the zygomatic arch and enhancing facial appearance. Specifically, the use of virtual planning of the surgery is a relatively new technique that allows for more precise pre-surgical planning compared to traditional methods. Additionally, assessing clinical matching in terms of orbital measurements and numerical models' registration is a more comprehensive and accurate way of evaluating the success of the surgical intervention. Previous studies have focused on one or the other, rather than com-

bining both. By utilizing both virtual planning and clinical matching assessment, this study provides a more holistic approach to reconstructing the zygomatic arch and improving the cosmetic outcome for patients with related facial injuries.

## 2. Case Presentation

A 31-year-old female with a previous right zygomatico-orbital complex fracture 3 years ago presented to the Department of Oral and Maxillofacial Surgery at Damascus University (Damascus, Syria). Her principal complaint was a secondary deformity resulting from the previous facial trauma. The clinical procedures adhered to the ethical standards established by the institutional and/or national research committee, as well as the 1964 Helsinki Declaration and its subsequent amendments or similar ethical standards. The Ethics Committee at Al-Andalus University for Medical Sciences granted approval for the study (No. 3 on 10 February 2022), with the ethics approval code AUEC-2022-02-10-003.

### 2.1. Study Design

This study outlines a method for numerical design and virtual planning using computer geometric design. The subject of the study was a 31-year-old female with a unilateral injury to the zygomatic complex. A TOSHIBA computerized tomography imager captured a series of tomographic images of the entire skull at a resolution of 0.25 mm. Mimics® software (version 21, Materialise, Leuven, Belgium) was used to import and segment the patient's skull CT images in DICOM format. This software isolated the skull tissues and converted them from 2D sections into an accurate 3D computer model that could be exported to other design and analysis programs as an STL format 3D surface structure. The study used 3-Matic® software (version 14, Materialise, Leuven, Belgium), a powerful tool for designing and optimizing complex structures from 3D scan data or CAD models. It allowed for the creation of parametric models that could be adjusted to meet surgical requirements. Surface smoothing, hole filling, and quality analysis were also performed using this software. Netfabb® software (version 2022, Autodesk, San Rafael, CA, USA) was used to compare between the models before and after the surgery and based on models' registration and assess model displacements, rotation, and similarity. This helps quantify the changes in orbital morphology and symmetry after the surgery.

### 2.2. Skull Model Generation

The most precise method to preserve the dimensions of a geometric structure without any changes or distortions is to construct a 3D computer model using computerized tomography scans (Figure 1). The 3D model of the compounding area was created through several stages. First, tomography images were imported in DICOM format, and the sections were validated as top/bottom, front/back, and right/left. Then, the required elements such as bone and soft tissues were separated from the rest using the Thresholding tool based on a specific greyscale called Hounsfield Units. The minimum threshold was set at 0 HU and the maximum at 1168 HU. The skull area was then identified, and the rest of the image was removed using a region-growing algorithm. Boolean operations were used to ensure that there were no overlaps, intersections, or gaps within the generated model before exporting it as a 3D object.

### 2.3. PSI Design

The PSI was designed using 3-Matic software, which allows the creation of parametric models that can be modified easily according to different specifications. The PSI was designed as a shell structure with a uniform thickness of 0.25 mm based on the mirroring of the normal side of the zygomatic complex (Figure 2). The mask-editing tool was used to virtually remove the old titanium plate from the right orbital CAD model. The left orbital CAD model was mirrored along a virtual median sagittal plane to obtain a symmetrical reference for the implant design. The mirrored orbital CAD model was subtracted from the right orbital CAD model to generate a preliminary PSI shape that matched the defect area.

The inner surface of the PSI was smoothed and reduced to provide a uniform gap between the implant and the bone. The locations for the fixation screws were determined based on the areas of bony support. Holes were created on the PSI to accommodate the screws. The final PSI design was exported as a Standard Triangle Language (STL) file, which is a format that only 3D printers can read.

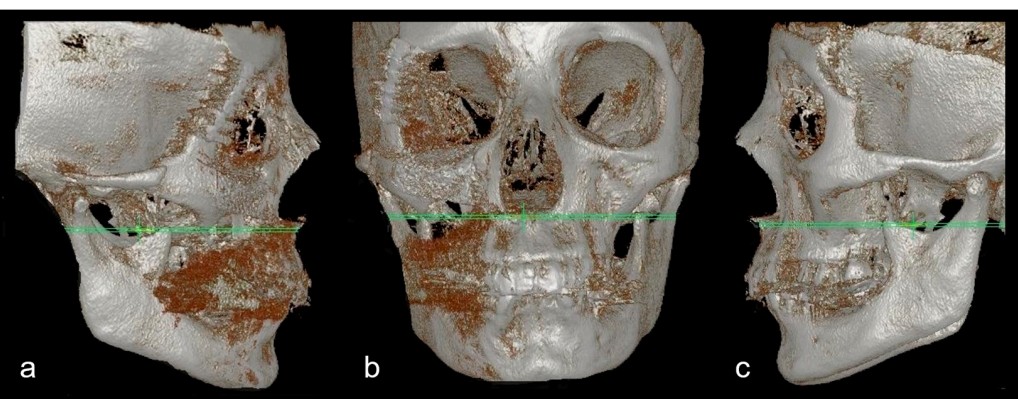

**Figure 1.** A 3D model of the skull before surgery by CT scan: (**a**). Lateral view of the injured side, (**b**). Anterior view, (**c**). Lateral view of the contralateral side.

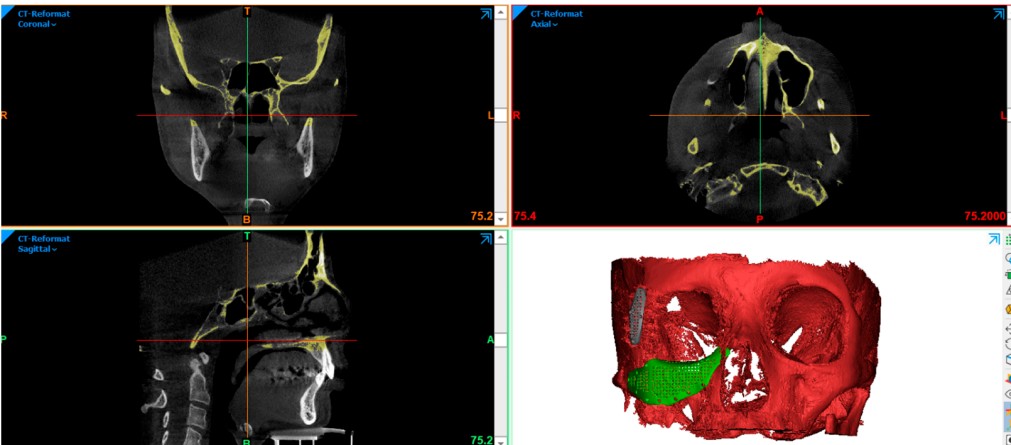

**Figure 2.** Overview of computed images on the three planes with a display of a 3D model built from the defined mask. Colored lines indicate the position of the cursor point placed on the 3D model on the three planes (coronal, sagittal, and axial).

### 2.4. Virtual Planning of Patients-Specific Implants

One of the challenges of orbital reconstruction is to achieve a precise fit of the implant to the defect and to restore the orbital contour. The implant was virtually attached to the right orbital bone using titanium screws through the holes in the implant (Figure 3).

The virtual planning enabled us to navigate the optimal method to expose the orbital rim and insert the implant into the defect. It also enabled us to determine the positions of titanium fixation screws and to verify the extent of orbital reconstruction and contour restoration.

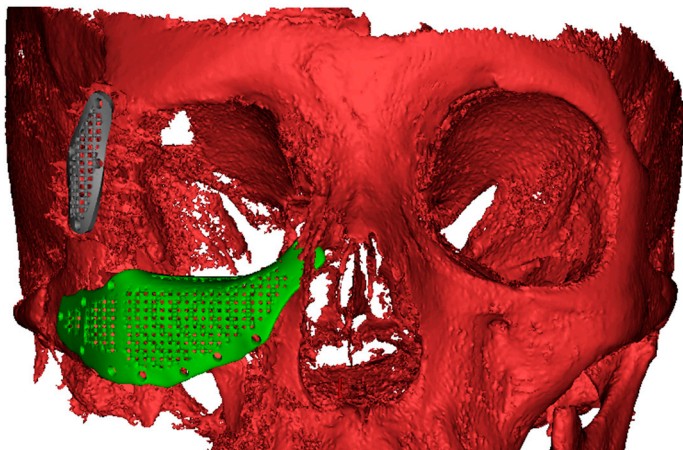

**Figure 3.** Virtual planning of the surgical intervention. The implant was divided into two parts: the upper part is shown in gray and the lower part is in green. The placement of two implants was verified before surgery and the screw locations were also confirmed to avoid interference with nerves or vessels in the area.

### 2.5. Manufacturing of Patient-Specific Implants and Surgical Procedure

A private laboratory manufactured the patient-specific implants using selective laser melting (SLM) technology with a Ti-6Al-4V alloy that had a periodic structure. The Riton D-150 printer from Guangzhou Riton Additive Technology Co., Ltd. (Guangzhou, China) was used to print the implants with an intended porosity of 85%, which the laboratory determined based on the printing steps and temperatures used. The powder layer thickness was 20 microns, and the printing accuracy was ±0.2%. The surgical approach was chosen by the surgical team (Figure 4). An infraorbital incision was done to expose the infraorbital edge. The infraorbital incision was made in the skin fold at the junction of the thin skin above the eyelid and the thicker skin above the cheek, parallelling the inferior orbital edge. The circular muscle was dissected at the same level, then an incision was made through the periosteum and lifted until exposing the infraorbital edge. The frontozygomatic suture was approached using a lateral eyebrow incision. The incision is made parallel to the hair of the eyebrow with a length of 2 cm. The periosteum was then cut and lifted laterally to reveal the lateral orbital margin. PSIs were fixed to the orbital rim with 2.2 mm screws. The timeframe for designing, manufacturing, and delivering patient-specific implants to our unit is 24 h.

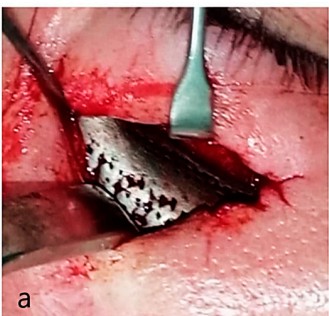
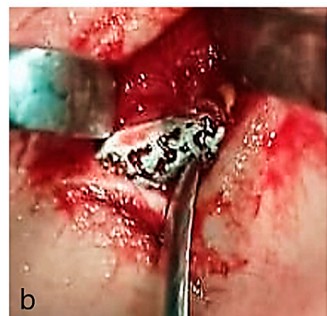
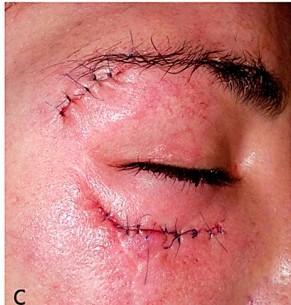

**Figure 4.** Surgical procedure: (**a**) wound exposure and insertion of the PSI, (**b**) the exact positioning of the PSI according to the virtual planning, and (**c**) final stitching.

### 2.6. Calculation of Orbital Measurements

It is important to perform some orbital measurements to evaluate the reconstructive effect of the surgery. We used two boundaries to define the orbital volume: the anterior orbital boundary and the posterior orbital boundary. The anterior orbital boundary is a straight line that connects the two points where the eye socket meets the face: the medial

orbital rim and the lateral orbital rim. The posterior orbital boundary is the point where the eye socket ends at the back of the skull: the orbital apex. Mimics software was used to make several orbital measures to assess the injury as shown in Figure 5.

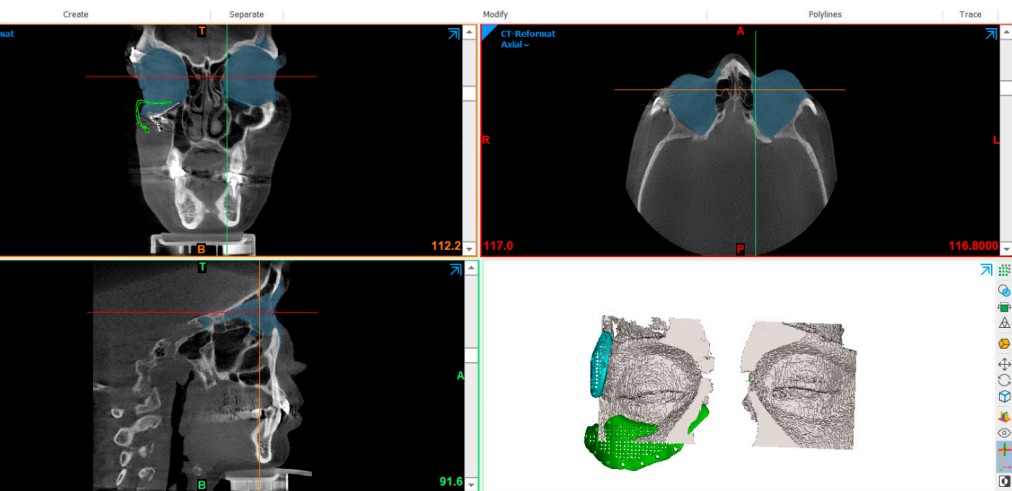

**Figure 5.** Measurement of orbital dimensions for fractured and normal orbits where the orbits were isolated and the 3D model was generated.

A statistical analysis of the orbital measurements were performed using SPSS software (version 27, IBM Corp., Armonk, NY, USA). Paired t-tests were applied to compare the values of the injured and normal sides before and after the surgery. We also calculated the percentage of improvement for each measurement by dividing the difference between the preoperative and postoperative values by the preoperative value. We considered a *p*-value of less than 0.05 to be statistically significant.

### 2.7. Models Registration

One of the challenges in reconstructive surgery is to measure the accuracy and effectiveness of the surgical outcome. Numerical models of the patient's anatomy were compared before and after the surgery using a registration technique. Netfabb software was used to compare the models based on mesh matching. A rigid registration algorithm was applied to align the models and compute the displacements between them. The displacements indicated the degree of change in the shape and position of the anatomical structures after the surgery.

## 3. Results

### 3.1. Post-Operative Clinical Evaluation

The clinical outcomes of the patient were observed for 6 months. There were no serious complications related to the surgery; the patient has not experienced wound breakdown, infection at the surgical site, or significant eye displacement or double vision that would affect their daily functioning or require additional intervention.

Based on the clinical evaluation of the patient's condition before and after the surgery (Figure 6a,b), the images indicated the necessity to restore the bone contour and symmetry between the eyes (Figure 6c). The outcomes revealed that the plate was accurately placed and adequately integrated with the bone tissue, resulting in the patient regaining the natural contour of her face with no vision complications. The results were satisfactory, and the patient resumed their routine activities without any difficulties or discomfort (Figure 6d).

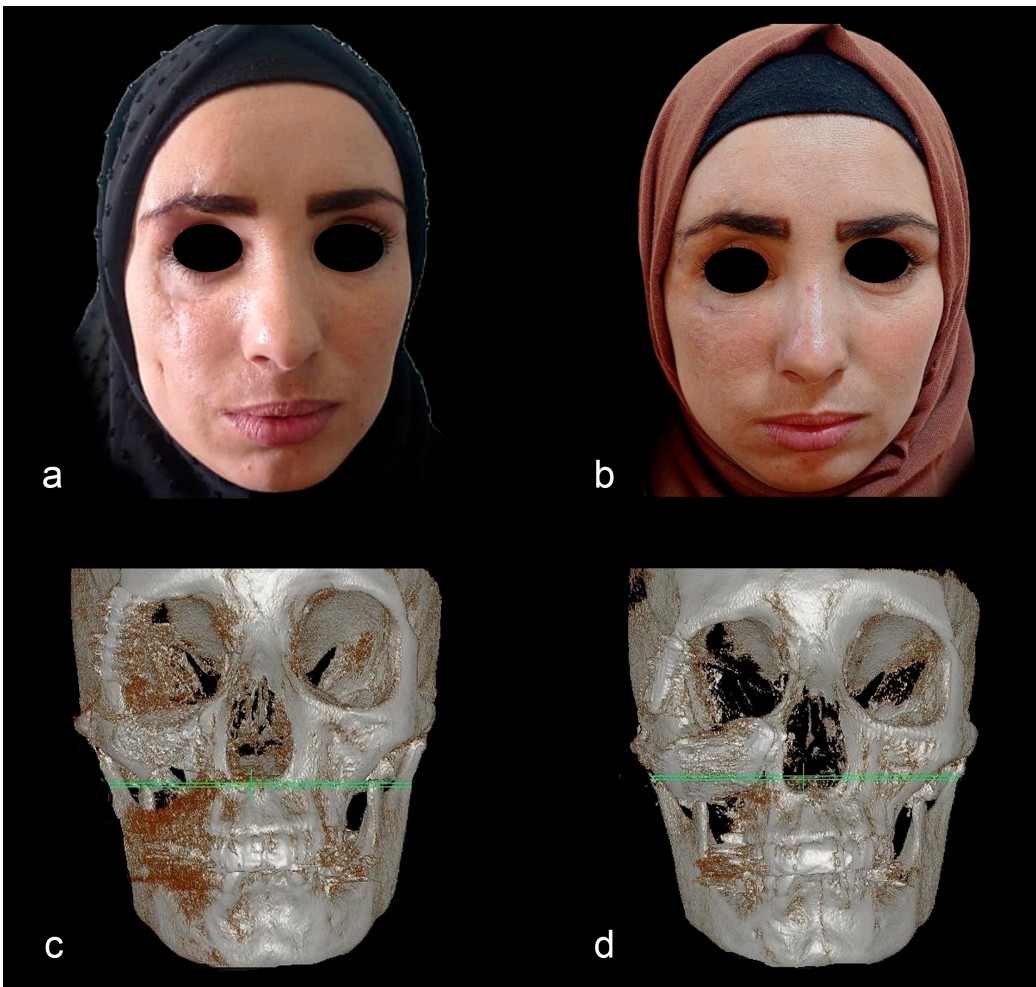

**Figure 6.** Clinical and radiological assessment before and after the surgery: (**a**) clinical situation before the surgery with the need for contour fixation after previous correction trials, (**b**) Favorable cosmetic outcome after the surgery with a minimal scar, (**c**) CT scan obtained before the surgical procedure, (**d**) CT scan of the patient after the surgery demonstrated that the implanted plates were appropriately positioned and appeared to have integrated well with the adjacent bone tissue.

*3.2. Pre- and Postoperative Orbital Measurements*

The results of fracture measurements are presented in Table 1. It is noticed that the injured side had higher orbital volume changes due to the fractures than the normal contralateral side (25,621.6 mm$^3$ vs. 26,727.1 mm$^3$) before the surgery.

**Table 1.** A comparison between orbital measurements before and after the surgery.

|  | Frontal Area [mm$^2$] | Perimeter [mm] | Depth [mm] | Volume [mm$^3$] |
|---|---|---|---|---|
| **Before Surgery** |  |  |  |  |
| Fractured | 1138.61 | 132.61 | 35.08 ± 2.53 | 25,621.6 |
| Contralateral | 1060.79 | 122.53 | 37.66 ± 1.67 | 26,727.1 |
| **After Surgery** |  |  |  |  |
| Fractured | 1142.32 | 125.62 | 36.22 ± 2.42 | 26,247.2 |
| Contralateral | 1082.30 | 122.53 | 37.66 ± 1.67 | 26,543.4 |

After the surgery, both sides seemed to have similar orbital volume values (26,247.2 mm$^3$ vs. 26,543 mm$^3$ for injured and contralateral sides, respectively) and zygomaticomaxillary depth difference (2.58 mm vs. 1.44 mm before and after the surgery, respectively). Orbital frontal

area measurements seemed to show significant improvement after the surgery (*p* < 0.001), as displayed in Table 2.

**Table 2.** A comparison between orbital measurements before and after the surgery. The symbol C represents the contralateral side and F the fractured side.

|  |  | Mean | Std. Deviation | Sig. (2-Tailed) |
|---|---|---|---|---|
| Pair 1 | VolumeC–VolumeF | 700.85000 | 467.24957 | 0.058 |
| Pair 2 | AreaC–AreaF | −68.92000 | 10.27683 | <0.001 |
| Pair 3 | PerimeterC–PerimeterF | −6.58500 | 4.03568 | 0.047 |

### 3.3. Evaluation of Clinical Matching

Models' registration study before and after the surgery showed a shape difference of 13.77%, indicating an 86.23% of reconstruction rate (Figure 7). This is a significant improvement compared to standard methods that achieved only 50% of the shape restoration rate on average.

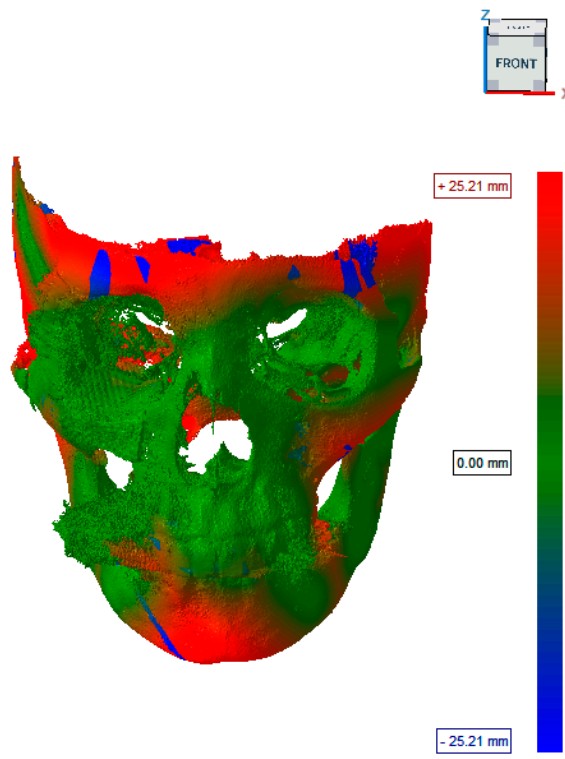

**Figure 7.** Clinical matching evaluation based on models' registration and comparison between numerical models before and after surgery. The results are shown on a color scale, where red represents positive displacements and blue represents negative displacements, measured in millimeters.

## 4. Discussion

The zygomatico-orbital complex is a critical anatomical region that affects both the function and appearance of the eye and face [22]. In this report, we describe a case of a patient who suffered a zygomatico-orbital complex fracture due to a traffic accident. The patient presented with orbital floor fracture, enophthalmos, diplopia, and facial asymmetry. She underwent two surgical procedures: one using the conventional approach and another using the contemporary approach. Through this case, we were able to compare the outcomes of both techniques and demonstrate the benefits of the contemporary approach over the conventional one. We suggest that the contemporary approach should be adopted as part of the standard protocol for treating zygomatico-orbital complex fractures.

Comminuted fractures are a type of bone fracture that occurs when a bone breaks into three or more pieces [23]. This usually happens due to severe trauma, such as a car accident or falling from a high place. Comminuted fractures can affect any bone in the body, but they are more common in long bones such as the femur, tibia, fibula, humerus, radius, and ulna [24]. One of the bones that can be affected by comminuted fractures is the zygomatic arch, which forms part of the cheekbone. It was found that most patients who suffer from comminuted fractures in this bone suffer from noticeable asymmetry, and about 70% of them have significant asymmetry [25]. This causes the patient's dissatisfaction with the results of the treatment provided [24,25].

The treatment of comminuted fractures in the zygomatic arch depends on several factors, such as the number and size of bone fragments, the degree of displacement and rotation, and the presence of other facial injuries. The main goal of treatment is to restore the normal anatomy and function of the zygomatic arch and to prevent complications such as infection, nerve damage, or malocclusion [26]. The most common treatment method is open reduction and internal fixation (ORIF), which involves making an incision near the fracture site and repositioning and stabilizing the bone fragments with plates and screws. ORIF can achieve good cosmetic and functional outcomes for most patients with comminuted fractures in the zygomatic arch. However, some patients may still experience residual asymmetry or deformity due to factors such as soft tissue swelling, scar formation, or inadequate fixation [27].

Another treatment method is closed reduction (CR), which involves manipulating and aligning the bone fragments without making an incision. CR can be done manually or with instruments, such as elevators or hooks. CR can be less invasive than ORIF and avoid potential complications such as infection or nerve injury. However, CR may not be suitable for patients with complex or unstable comminuted fractures in the zygomatic arch that require precise fixation [28]. Salma et al. mentioned the diplopia management during complex zygomatic fractures [29]. Sometimes, a second surgery may be needed to address functional vision problems, such as enophthalmos and persistent diplopia. Furthermore, some patients may opt for revision surgery to improve their cosmetic appearance only.

The extent of bone fragmentation and bone loss caused by the injury is a major factor that influences the functional and aesthetic outcomes, as it affects the soft tissues, facial contour support, and orbital framework. Feng et al. first suggested the mirror technique in restoring the symmetry and function of the injured zygomatico-orbital complex [30]. This technique involves creating a three-dimensional model of the unaffected side and using it as a template to guide the reduction and fixation of the fractured bones. Our results of plate stability and shape recovery reinforce this principle. The mirror technique has advantages over conventional methods, such as reducing operative time, minimizing blood loss, improving aesthetic outcomes, and avoiding donor site morbidity.

The strengths of this study reside in evaluating the use of patient-specific implants for the reconstruction of complex zygomatico-orbital fractures. This is based on quantitative indicators including volume, area, and depth of the internal orbit in a patient who underwent surgery with patient-specific implants. The results showed that a high restoration rate based on the models' registration before and after the surgery. This study had several limitations. The suggested methodology performs the model's superimposition and comparison of the whole skull models, which is one limitation of this shape similarity estimation. An automatic isolation and comparison of orbit models would produce more precise results. Moreover, the study needs to be applied to an adequate number of cases to confirm the average needed time for design and production and also to follow both the clinical and stability outcomes of the reconstructive surgery.

## 5. Conclusions

Titanium patient-specific implants (PSIs) are an innovative approach that seemed to enhance the well-being of patients with complicated orbital fractures. By using the computerized technique, we were able to create the plates with high precision and evaluate

the best locations for fixation. The patient expressed a positive outcome in terms of cosmetic and functional aspects. Considering the study's limitations, further research is needed to explore the mechanical properties and biocompatibility of various titanium alloys, as well as the impact of pore size and porosity rate on the bone integration and stability of the implants. Additionally, more clinical trials are required to contrast titanium PSIs with other methods of orbital reconstruction in terms of functional and aesthetic outcomes, cost-efficiency, and patient satisfaction.

**Author Contributions:** Conceptualization, methodology, data curation, software: M.A.D. and H.M.N.; investigation, visualization, formal analysis, writing—original draft: K.Y.; supervision, project administration, validation: K.D. and G.A.; resources, writing—review and editing: A.I. All authors have read and agreed to the published version of the manuscript.

**Funding:** This research received no external funding.

**Institutional Review Board Statement:** The study was conducted according to the guidelines of the Declaration of Helsinki and approved by the Ethics Committee at Al-Andalus University for medical sciences (No. 3 on 10 February 2022).

**Informed Consent Statement:** The patient provided an informed consent statement.

**Data Availability Statement:** Data are available from the corresponding author upon reasonable request.

**Conflicts of Interest:** The authors declare no conflict of interest.

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
