# Peer review of "A Case Report of Zygomatic Fracture Reconstruction: Evaluation with Orbital Measurements and Models Registration"

_applsci, doi:10.3390/app13106154_

Round 1

Reviewer 1 Report

Dear Authors, 

Congratulations on the work you have done and presented in this manuscript. I believe your work can be published after some major modification to the text. Please see the attachment. 

Author Response

First of all, we would like to thank you for reading and reviewing our paper.
We did our best to address all the comments. Moreover, major revisions have been made to our manuscript, taking into consideration the following modifications below.

Reviewer 2 Report

The authors reported a case of zygomatic and maxillary fracture treated with personalized implant. There is no doubt that a good therapeutic effect has been achieved . However, there are still some issues which should be addressed.

1. It should be clearly stated in the title the manuscript is a case report.

2. The abstract is too vague and needs to be rewritten. Please refer to the format of other Case report.

3. There have been many reports on the reconstruction of zygomaticomaxillary complex with personalized implant. What are the problems and shortcomings of the previous studies? What is the author's innovation?

4. Preoperative photos of the patient should be provided for comparison with postoperative photos.

5. The figures are not clear enough, it is suggested to replace them with some clearer figures.

6. The implant comes from A private laboratory, is it clinically approved for use? Is this study approved by the Ethics Committee?

7. Is the use of patient pictures approved?

8. The implant is similar to titanium mesh, under which a dead cavity will be formed, which is prone to infection. How to solve the problem?

9. Why wasn't the implant designed as a solid instead of a mesh?

10. Bilateral symmetry should be measured.

So, major revision should be recommended.

Author Response

(The authors gave the same response as above.)

Round 2

Reviewer 1 Report

Thank you for addressing all the questions that I have raised. I have no further remarks.

Reviewer 2 Report

The revised is much better than the original version. I have no other questions. Acceptance should be recommended for this manuscript.